# Mature and Immature/Activated Cells Fractionation: Time for a Paradigm Shift in Differential Leucocyte Count Reporting?

**DOI:** 10.3390/diagnostics11060922

**Published:** 2021-05-21

**Authors:** Rana Zeeshan Haider, Najeed Ahmed Khan, Eloisa Urrechaga, Tahir Sultan Shamsi

**Affiliations:** 1Baqai Institute of Hematology, Baqai Medical University, Karachi 75340, Pakistan; 2National Institute of Blood Disease (NIBD), Karachi 75300, Pakistan; t.shamsi.62@gmail.com; 3Department of Computer Science, NED University of Engineering and Technology, Karachi 75270, Pakistan; najeed@neduet.edu.pk; 4Core Laboratory, Galdakao-Usansolo Hospital, 48960 Galdakao, Spain; eloisamaria.urrechagaigartua@osakidetza.eus

**Keywords:** CBC, extended differential leucocyte count, NEUT#&, LYMP#&, IG, HFLC, hematology analyzer

## Abstract

Leucocytes, especially neutrophils featuring pro- and anti-cancerous characteristics, are involved in nearly every stage of tumorigenesis. Phenotypic and functional differences among mature and immature neutrophil fractions are well reported, and their correlation with tumor progression and therapy has emerging implications in modern oncology practices. Technological advancements enabled modern hematology analyzers to generate extended information (research parameters) during complete blood cell count (CBC) analysis. We hypothesized that neutrophil and lymphocyte fractions-related extended differential leucocytes count (DLC) parameters hold superior diagnostic utility over routine modalities. The present study was carried out over a four-and-a-half-year period wherein extended neutrophil (immature granulocyte [IG] and mature neutrophil [NEUT#&]), and lymphocyte (activated/high fluorescence lymphocyte count [HFLC] and resting lymphocyte [LYMP#&]) parameters were challenged over routine neutrophil [NEUT#] and lymphocyte [LYMP#] items in a study population of 1067 hematological neoplasm patients. Extending the classical statistical approaches, machine-learning-backed data visualization was used to explore trends in the study parameters. As a whole, extended neutrophil and lymphocyte count outperformed and was diagnostically more relevant than routine neutrophil and lymphocyte parameters by showing the least difference from their respective (gold-standard) manual DLC counts. The mature neutrophil count was compared to IG, and resting lymphocyte count was compared to HFLC by calling the function ‘correlation’ as a ‘clustering function’ for heatmap based visualization. The aforementioned study parameters displayed close clustering (rearrangement) for their respective study items by presenting distinct trends of equally valuable weights (deviated values), advocating fractions-based extended DLC reporting. Importantly, using a Bland and Altman analysis analogously to a manual neutrophil count, the mature neutrophil count [NEUT#&] remained unbiased since a routine neutrophil count [NEUT#] was found to be a negatively biased. The extended DLC-parameter-driven fractions-based reporting has superior diagnostic utility over classical routine approaches; this finding can largely minimize labor-intensive manual DLC practices, especially in hematology–oncology departments.

## 1. Introduction

Complete blood cell counting (CBC) and differential leucocyte counting (DLC) are baseline indices routinely used in clinical work. The indications for CBC are numerous, but they are mainly infectious and hematological disorders. The first choice of information for clinicians in a DLC is neutrophil count, followed by eosinophil, monocyte, lymphocyte and basophil [1]. Despite the fact that neutrophils and lymphocytes contribute 50–70% and 20–30% respectively of total peripheral leucocytes, they just recently received pathophysiologic and therapeutic attention for cancers [2]. Neutrophils are reported to have a wide range of pro- and antitumor activities, including neoplastic cell proliferation, direct tumor cell killing, metastasis, angiogenesis and triggering related immune responses [3,4]. The key role of lymphocytes in anti-cancerous response by encouraging apoptosis and through restraining the migration and proliferation of cancerous cells is well studied [5,6,7]. Mature and immature neutrophils differ in their functional and phenotypic capacities with regard to tumor progression and the efficacy of tumor therapy [8]. Activated and proliferating lymphocytes functionally contrast with mature and resting lymphocytes in the inhibition of neoplastic cell migration/proliferation and cytotoxic cell death that can be a potential therapeutic monitoring marker for neoplasms [9]. Therefore, it is necessary to recognize immature and mature neutrophils or lymphocytes in first-line tests (CBC), and extended fractions should be individually interpreted with a DLC report.

Technological advancements in modern hematological analyzers make it possible to generate a broader picture of hematological parameters CBC and DLC testing [10]. DLC analysis features the extended enumeration of leucocyte fractions (mature, immature, active/reactive), which include handy parameters (counts) for the immature granulocyte (IG) and total neutrophil counts (NEUT#); likewise, hyperactive/reactive lymphocytes (high fluorescence lymphocyte count (HFLC)) are separated from total lymphocyte count (LYMP#) to generate extended (only mature) neutrophil (NEUT#&) and (only resting) lymphocyte (LYMP#&) counts. Automated IG counting has been widely reported to be a useful clinical tool for the detection, monitoring and progression of inflammation [11], sepsis [12,13,14,15] and bone-marrow neoplasms [16]. Moreover, a few studies have also reported HFLC as a potential predictor for the prediction of peripheral plasma cells [17], atypical lymphocyte cells [18], septicemia and viral (dengue) fever [19,20]. However, notably, extended neutrophil and lymphocyte counts are never challenged over classical neutrophil (neutrophil plus IG) and lymphocyte (lymphocyte plus HFLC) parameters. In many cases, clinicians’ decisions to ‘treat’ or ‘not’ are principally based on absolute counts of neutrophils and lymphocytes. The present study was conducted to evaluate the potential diagnostic superiority of extended neutrophil and lymphocyte counts over classical routine neutrophil and lymphocyte parameters among a study population with common hematological neoplasms. Gold-standard manual peripheral blood-film DLC counts were used as reference parameters.

## 2. Materials and Methods

Modern hematology analyzers possess the option to automatically generate counts for WBC’s mature and immature fraction as immature granulocytes (IG) and hyperactive/reactive lymphocytes (HFLC) in peripheral blood samples using different methodologies under the flag of extended DLC items. The Sysmex XN series by using a specific lysing agent (Lysercell WDF) and fluorescence dye for RNA/DNA content (Fluorocell WDF reagent) is capable to generate extended DLC items including IG and HFLC Three principle measurements: for size (forward scatter), for granularity/cytoplasmic complexity (side scatter) and for RNA/DNA content (side fluorescence light), are taken for differentiation and the counting of these extended DLC items in a designated analytic channel named ‘white blood cell differential (WDF)’. In parallel to the virtual presentation of these items on the WDF channel backend scattergrams, these advance hematology analyzers also generate quantitative values (numbers) for extended DLC parameters. The clusters (scattering area) of IG and HFLC can be noticed on the upper edges of the neutrophil and lymphocyte scattering areas, respectively. While the parametric values for IG and HFLC are displayed under ‘DLC Research Items’ option in analyzer software and have the option to moved them within a working window of the routine classical CBC and DLC parameters. Subsequently, new mature neutrophil (NEUT#&) and resting lymphocyte (LYMPH#&) counts are auto-calculated through the deduction of IG and HFLC values from classical routine neutrophil (NEUT#) and lymphocyte (LYMPH#) counts and displayed along with other extended DLC items in the research parameter window.

In the current round of study, the NEUT#& and LYMH#& from automated extended DLC parameters were challenged over NEUT# and LYMP# among the automated classical DLC items in designation of a manual peripheral blood film based neutrophil and lymphocyte counts (from 500 WBC DLC) as a gold standard. A total of 1067 patients with 181 acute myeloid leukemia (AML) (excluding acute promyelocytic leukemia (PML)), 44 APML (PML-RARA), 89 chronic myeloid leukemia (CML), 51 myelodysplastic syndrome (MDS), 71 myeloproliferative disorders (MPN) except CML, 10 MDS/MPN, 136 acute lymphocytic leukemia (ALL), 9 Hodgkin’s lymphoma (HL), 95 non-Hodgkin’s lymphoma (NHL), 32 multiple myeloma (MM) and 349 normal control patients were recruited for the present study over a period of four and half years (January 2014 to June 2018). Blood samples were received from our inpatient and outpatient departments at the Hematology Section of the Central Diagnostic and Research Laboratory, National Institute of Blood Disease and Bone Marrow Transplantation (NIBD & BMT), Karachi-Pakistan. Blood collection was conducted in EDTA (dipotassium ethylenediaminetetraacetic acid) in a purple top (BD, plastic whole blood tube with spray-coated K2EDTA, 3.0ml) during the diagnostic workup of study cases. Purple-top specimens were analyzed for detailed complete blood count (CBC) parameters using the extended differential leucocyte count (DLC) mode on the XN-1000 Sysmex (Co., Kobe, Japan) within four hours of blood collection. As per instructions provided by the manufacturer (Sysmex), the analyzer was used in its routine mode. All CBC and DLC items were reported after qualifying at all three levels of internal quality control (IQC) and the RIQAS external quality assessment (EQA) scheme for CBC testing.

Aiming at the assessment of intra-assay reproducibility, four samples with varying extended DLC counts were selected. Each sample was performed ten times in the automatic routine mode. Next, the percent coefficient of variation (%CV) was computed by using equation ‘%CV= (SD/mean) × 100’.

Simultaneously, peripheral blood films were examined for a manual DLC count of at least 500 leucocytes, performed independently by two experienced hematologists. The morphologists that performed DLCs were masked to the DLCs determined by the other morphologists and by the automated DLC. Next, the DLC of both morphologists were averaged and absolute values were used. The DLC was performed using criterion defined by the College of American Pathologists [21].

Data was analyzed using SPSS version 23.0 and visualized through Clustvis: a web tool for visualizing the clustering of multivariate data (inspired by the PREDECT project and mostly based on BoxPlotR codes). The median, along with the interquartile range (IQR), was used for continuous variables. One sample *t*-test was conducted to explore the trends among study parameters, and mean difference with standard deviation (S.D.) along significant value (*p*-value) was reported. A Bland and Altman analysis was inducted to assess biasness and agreement between selected study parameters at a predefined confidence interval of 95%. In addition, a heat map (a supervised data visualization tool) was used to delve into and visualize the subtle patterns of study parameters among study groups. In the clustering of study parameters, the ‘correlation’, ‘average’ and ‘tightest cluster first’ functions were called the ‘clustering distance’, ‘clustering methods’ and ‘tree ordering for rows’, respectively. The heat map color scheme ‘diverging: RdBu’ at ‘minimum -2 to maximum 2′ heat map color range was called for color grading because it contains diverging palette options more suitable for data with both negative and positive values, as in our case.

The institutional review board of an academic research center (NIBD and BMT) approved this study with the permit number: NIBD/RD-167/14-2014.

## 3. Results

The routine analyzer generated CBC-reporting parameters among our study groups, pictorially illustrated through the heat map in Figure 1. The heat map illustration not only color-grades the parameters (rows) to assist a quick view of hot and cold spots within the table (dataset) but also clusters (rearranges) study parameters (rows) or groups (columns) with identical patterns by nodeing (branching) them. In our case, we applied clustering for rows (study parameters) only. As a whole, cold spots (lower values) were noted for RBCs, Hb and platelets while neutrophils, eosinophils, basophils, WBCs, and NRBCs presented hot spots. Notably, lymphocyte and monocytes showed a mixed pattern (both cold and hot spots). The parameters with cold spots and hot spots were clustered (node up) on the upper and lower areas of the heat map, respectively. At the same time, parameters with mixed patterns were node up in the middle of the heat map. The nodeing trends help us to find how closely patterned to each other our study parameters are. The step/level of any particular node where it groups to other node/s describes its degree of clustering (correlation, in our case). The first step nodeing was observed between eosinophils, basophils and neutrophils, and WBCs and NRBCs. The aforementioned parameters’ nodes cluster up to each others’ nodes at second level Monocytes join this node at the third step, while in the upper area, Hb and platelets also node up at the same step. Lymphocytes and RBCs followed the trend of fourth-step nodeing and clustered to the major nodes of monocytes and Hb, respectively. The lower the level of nodeing, the closer the values of the study parameters.

The neutrophil and lymphocyte extended DLC items on the heat map presentation in general followed identical trends to their corresponding items (IG and HFLC, respectively), with few exceptions (Figure 2). Regarding nodeing trends, first- and second-step clustering were noted for neutrophil and IG, and for lymphocyte and HFLC, respectively. This indicates that neutrophil-related items have closer behavior to each other but keep equally valuable weight (higher values) that can’t be overlooked. From the initial analysis, among neutrophil and lymphocyte extended DLC items, the mature neutrophil count superiorly challenged its classical (routine) DLC item.

As modern hematology analyzers have limitations beyond the six-part DLC, it was necessary to perform manual DLC on peripheral blood films to explore the types and counts of immature/abnormal blood cells that were potentially over-fitted (falsely counted) in the IG and HFLC counts. As shown in Table 1, the most common immature/abnormal WBCs types noted in our study group were myelocytes, metamyelocytes, blast cells and abnormal lymphoid cells. Each of the aforementioned WBCs has its own diagnostic utility and clinically demands its separation from mature neutrophils and lymphocytes, at least if automated individual reporting is not possible.

To assess the potentially superior (significant) diagnostic trend between the mature neutrophil count over its classical parameters in respect to its gold standard, manual neutrophil count, one sample *t*-test was conducted. In this analysis, a classical neutrophil item showed a mean difference of −7.79 with 37.03 S.D. at significant value (*p*-value: <0.001). It failed to prove any significant level of agreement for manual neutrophil count. At the same time, the mature neutrophil count, by giving its mean difference of 1.36 with 19.01 S.D. at statistically insignificant value (*p*-value 0.056), demonstrated its significant level of agreement with the manual neutrophil count. Furthermore, the Bland and Altman analysis was used to affirm the above-mentioned exploratory findings. The Bland and Altman plot showed the systematically negative bias on classical items (NEUT#), while the extended parameters (NEUT#&) remained unbiased (Figure 3), which upheld the diagnostic reporting advantage of the mature neutrophil count in comparison to the classical neutrophil count.

## 4. Discussion

The early prediction and differentiation of immature/abnormal blood cells from their mature forms, particularly for WBCs remains key CBC/DLC reporting challenge. It could become more diagnostically exacting at hematology–oncology clinics and diagnostic laboratories with heavy CBC plus DLC testing. Therefore, the introduction of modern hematology analyzers generated extended CBC and DLC parameters is potentially a valuable perspective especially on hematological diagnostic end.

En masse, the results of the present study show that, in comparison to that of extended DLC (neutrophil and lymphocyte), the parameters showed a superior level of agreement over classical (routine) items to standard manual DLC items (Figure 3) and can also successfully flag the presence of immature/abnormal blood cells (Table 1). In this challenge, DLC extended items versus classical parameters, the mature neutrophil item showed greater potential. Additional key evidence emerged from the present study, showing that CBC and DLC reporting is likely to become more comprehensive by replacing classical (and limited) DLC items with suggested extended parameters. According to our knowledge, this is the first study to suggest the replacement of routine DLC lymphocyte and neutrophil items with extended parameters in CBC plus DLC reporting. However, the related findings, including the reporting of automated IGs by replacing the manual IG count, are in agreement with reports available from the literature [22]. IG values might be exceptionally informative, not only at the diagnosis stage, but also during follow-ups, as IG values were reported to consecutively increase pursuant to the severity of the disease [23]. Nevertheless, as the automated IG has its own limitations, including false counting of other abnormal/immature blood cells, not only band cells, myelocytes and metamyelocytes, the complete replacement of manual IG counting, particularly in hematology/oncology laboratories, is not advisable [24]. It can be suggested that the better approach is to use an automated IG as a flagging (screening) parameter for immature granulocytes that may be followed by a manual review of the peripheral film. The diagnostic utility of HFLC was also investigated and reported to be a flagging item for lymphocytes/lymphoid cells with high RNA content, including atypical lymphocytes, plasma cells and others [13,18,25]

Interestingly, the values of mature neutrophil count, IG, resting lymphocyte count and HFLC displayed nonpartisan and peculiar trends (Figure 2), and the diagnostic impact will be contingent on the addition of these DLC details in the CBC report. Although, these extended DLC parameters are generated in routine CBC plus DLC mode without additional reagent use, most are still not used in practice by most clinics. The missing external-assurance scheme and specific reference ranges limit their diagnostic applicability. The test characteristics of these items should be further explored through extended studies to exemplify the diagnostic value of the concerned parameters for the prediction of specific clinical disorders.

## 5. Conclusions

In conclusion, the diagnostic superiority of the extended DLC items at an acceptable level of agreement with gold-standard manual DLC parameters is advised. Furthermore, the reporting of these extended DLC details is promising for the diagnostic needs of hematology–oncology clinics, and therefore, the ordering of manual DLC may be needed in a few cases.

## Figures and Tables

**Figure 1 diagnostics-11-00922-f001:**
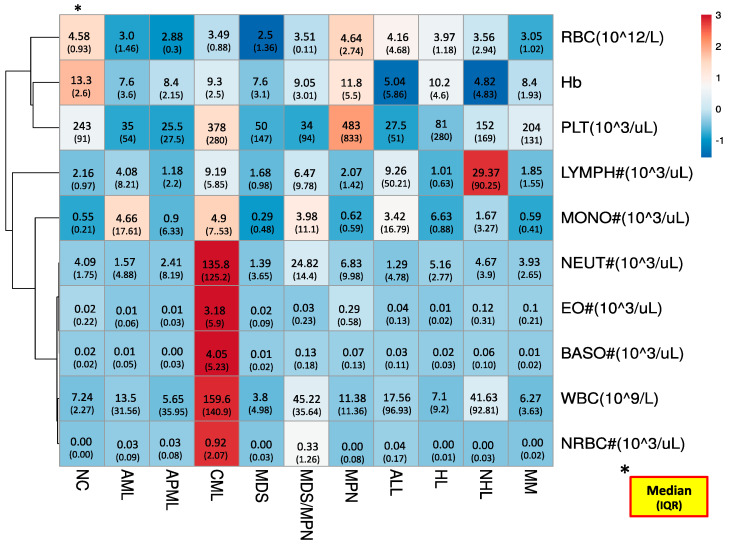
The heat map: color grading and clustering trends of routine CBC reporting parameters among study groups. Red color is used for higher values, and for lower values, blue color is used. Hb: hemoglobin; RBC: red blood cell; WBC: white blood cell; PLT: platelet; NEUT#: absolute neutrophil count; LYMP#: absolute lymphocyte count; MONO#: absolute monocyte count; EO#: absolute eosinophil count; BASO#: absolute basophil count; NRBC: absolute nucleated RBC; IQR: interquartile range. * The statistics of study items are presented as ‘Median (IQR)’ fashion.

**Figure 2 diagnostics-11-00922-f002:**
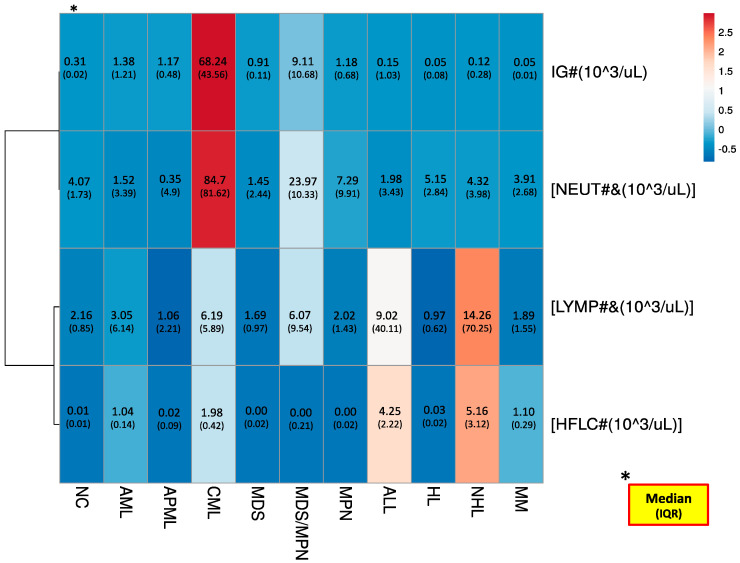
Heat map presentation of the extended DLC items (by median (IQR statistics) among study groups). IG#: absolute immature granulocyte count; HFLC#: absolute high fluorescence lymphocyte count; NEUT#&: minus IG absolute neutrophil count; LYMPH#&: minus HFLC absolute lymphocyte count. * The statistics of study items are presented as ‘Median (IQR)’ fashion.

**Figure 3 diagnostics-11-00922-f003:**
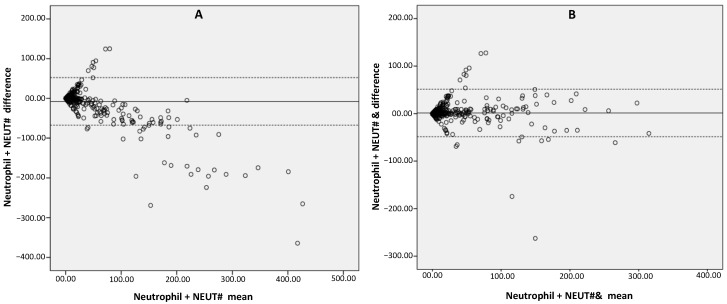
Bland and Altman plot comparing (**A**) the classical neutrophil count (NEUT#) and (**B**) the mature neutrophil count (NEUT#&) against the manual neutrophil count (Neutrophil). The mean absolute difference is shown by the central line while upper and lower dotted lines are designated for the 95% CI of the mean difference.

**Table 1 diagnostics-11-00922-t001:** Peripheral blood film differential leucocytes count among study groups.

Manual DLC Items	Control	AML	APML	CML	MDS	MDS/MPN	MPN	ALL	HL	NHL	MM
Neutrophil	4.09 (1.75)	1.62 (6.94)	0.23 (1.687)	86.18 (18.32)	1.48 (1.59)	26.68 (6.06)	7.97 (2.38)	2.28 (24.23)	5.61 (0.74)	7.91 (24.13)	4.04 (0.85)
Lymphocyte	2.16 (0.97)	2.02 (9.15)	1.02 (12.46)	4.79 (7.05)	1.75 (2.04)	5.43 (4.28)	1.93 (2.04)	3.42 (29.08)	0.92 (1.56)	5.41 (37.12)	1.82 (0.63)
Monocyte	0.55 (0.21)	0 (0.94)	0 (0.52)		0.11 (0.25)	0.90 (4.98)	0.34 (0.34)		0.49 (0.74)	0.42 (2.78)	0.31 (0.14)
Eosinophil	0.2 (0.22)			4.79 (4.23)	0 (0.10)	0 (0.36)	0.23 (0.34)				0.06 (0.11)
Basophil	0.02 (0.02)			6.384 (7.05)		0 (0.36)	0.11 (0.23)				
Myelocyte		0 (0.63)	0 (0.26)	35.11 (18.32)	0.04 (0.15)	2.71 (3.21)	0 (0.23)		0 (0.18)		0 (0.05)
Metamyelocyte				7.98 (8.46)	0 (0.05)	1.36 (2.14)					
Promyelocytes				0 (1.41)							
Blast		5.67 (20.51)		3.19 (2.81)		2.03 (2.85)		7.29 (64.94)			
Abnormal Promyelocyte			2.99 (20.37)								
Abnormal lymphoid cell										24.14 (73.32)	

## Data Availability

The codes and data studied are accessible at National Institute of Blood Disease (NIBD) Research Database, condition to a request matching NIBD Research Ethics.

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
