# Peer review of "Mature and Immature/Activated Cells Fractionation: Time for a Paradigm Shift in Differential Leucocyte Count Reporting?"

_diagnostics, 2021, doi:10.3390/diagnostics11060922_

Round 1
Reviewer 1 Report
In my opinion, the work is written in unclear English. Large parts of the article, including the title, are difficult to understand. I found no new data in the manuscript. The described parameters are generally known and commonly used in the routine diagnosis of hematological neoplasms. The research objectives are not precise and insufficiently explained. The results are presented in an unclear manner, and the lack of information on the analysis of the data distribution makes them difficult to interpret. It is worth remembering that the results expressed as mean and standard deviation are characteristic of data with a normal distribution, and for data with a distribution other than normal, the median and IQR should be provided. The discussion is misses the point. It should focus on the context, answer the question and hipotheses raised in introduction, and discuss the advantages and disadvantages or limitation of this study critically.
Author Response
Thank you very much for raising these important points. Following is our point-by-point response;
- General English of our work was focused and tried to use quality English.
- For better understanding of our article, including title almost every sentence is tried to change/modify or corrected.
- Presentation and description of our study parameters are made more-easy to understand and their novelty is explained.
- After providing some background, in last of introduction section we tried to precisely describe explained our study objectives.
- Results section is specially focused and new R based machine learning driven data visualization tool (Heat map) is used to explore substantial trends in our data. With color grading, clustering of parameters with identical trends is performed.
- As per suggestion, Median and IQR are provided.
- Discussion is also extensively modified to address all key elements of hypothesis/introduction and results of our study. Limitation of present study is also tried to address.
Reviewer 2 Report
The study explores the potential superiority of extended differential leucocyte count over automated methods, in comparison to manual peripheral blood count, for the diagnosis of a variety of haematological neoplasms.
The study design and population seem appropriate.
However, the text is not fluent and the reading is very difficult, mainly because of the high number of acronyms, which have been largely used for describing both blood sub-populations and haematological malignancies.
I would suggest the authors to include at least a graphical resume of the blood cells analysed.
An extensive editing of the English language is also recommended to ease the reading.
Author Response
Thanks for kind comments. The whole article is rewritten/modified/corrected with quality English. The uses of acronyms are avoided.
Reviewer 3 Report
The authors describe the utility of a new hematology analyzer for reporting the results of CBC. Neutrophil counts could be calculated after removing immature neutrophils. Similarly, lymphocytes are counted after the removal of immature lymphocytes. The study included 1067 patients over 4.5 years. The study enrolled many patients with hematological malignancies. In their conclusion, the new method was not superior to the classical way of reporting CBC.
The manuscript is difficult to read. The abstract introduced three new terms: " extended neutrophil counts," " classical neutrophil counts," and "standard neutrophil counts." Please, define any new term before using it in the abstract to make it easier to read.
Few typos and grammar problems need attention. For example, on page 3, "tabe 2" means "table 2."
Author Response
Thank you for raising key comments. The extensive modification with quality English is completed to make our work more understandable. The similar words are tried to renamed or used in more clear manners. Typos and grammar problems are tried to sort out.
Round 2
Reviewer 1 Report
Thank you for all the amendments made (changing the title is also a very good procedure), for precisely describing the purpose and for any explanations. While most of the aspects discussed are known as they are, the manuscript is acceptable for publication.
One explanation: The quality of the English language is good. I meant the unclear formulation of the aim of the work, the description of the results and discussion.
Reviewer 3 Report
The manuscript has improved.